# Acute Exercise with Moderate Hypoxia Reduces Arterial Oxygen Saturation and Cerebral Oxygenation without Affecting Hemodynamics in Physically Active Males

**DOI:** 10.3390/ijerph19084558

**Published:** 2022-04-10

**Authors:** Gabriele Mulliri, Sara Magnani, Silvana Roberto, Giovanna Ghiani, Fabio Sechi, Massimo Fanni, Elisabetta Marini, Silvia Stagi, Ylenia Lai, Andrea Rinaldi, Raffaella Isola, Romina Vargiu, Marty D. Spranger, Antonio Crisafulli

**Affiliations:** 1Department of Medical Sciences and Public Health, University of Cagliari (Italy), 09124 Cagliari, Italy; jabutele84@gmail.com (G.M.); saretta84.m@gmail.com (S.M.); silvy_rob@yahoo.it (S.R.); giovanna.ghiani@tiscali.it (G.G.); fabiosechiaho@gmail.com (F.S.); max_f.93@hotmail.it (M.F.); 2International PhD in Innovation Sciences and Technologies, University of Cagliari (Italy), 09124 Cagliari, Italy; 3Department of Life and Environmental Sciences, University of Cagliari (Italy), 09124 Cagliari, Italy; emarini@unica.it (E.M.); silviastagi89@gmail.com (S.S.); 4Department of Biomedical Sciences, University of Cagliari (Italy), 09124 Cagliari, Italy; ylenia.lai@gmail.com (Y.L.); rinaldi@unica.it (A.R.); isola@unica.it (R.I.); rvargiu@unica.it (R.V.); 5Department of Physiology, Michigan State University, East Lansing, MI 48823, USA; mds@msu.edu

**Keywords:** blood pressure, cardiac output, stroke volume, ventricular filling rate, ventricular emptying rate

## Abstract

Hemodynamic changes during exercise in acute hypoxia (AH) have not been completely elucidated. The present study aimed to investigate hemodynamics during an acute bout of mild, dynamic exercise during moderate normobaric AH. Twenty-two physically active, healthy males (average age; range 23–40 years) completed a cardiopulmonary test on a cycle ergometer to determine their maximum workload (W_max_). On separate days, participants performed two randomly assigned exercise tests (three minutes pedaling at 30% of W_max_): (1) during normoxia (NORMO), and (2) during normobaric AH at 13.5% inspired oxygen (HYPO). Hemodynamics were assessed with impedance cardiography, and peripheral arterial oxygen saturation (SatO_2_) and cerebral oxygenation (Cox) were measured by near-infrared spectroscopy. Hemodynamic responses (heart rate, stroke volume, cardiac output, mean arterial blood pressure, ventricular emptying rate, and ventricular filling rate) were not any different between NORMO and HYPO. However, the HYPO test significantly reduced both SatO_2_ (96.6 ± 3.3 vs. 83.0 ± 4.5%) and Cox (71.0 ± 6.6 vs. 62.8 ± 7.4 A.U.) when compared to the NORMO test. We conclude that an acute bout of mild exercise during acute moderate normobaric hypoxia does not induce significant changes in hemodynamics, although it can cause significant reductions in SatO_2_ and Cox.

## 1. Introduction

Athletes often use acclimatization at high altitude to improve performance at sea level [1,2,3,4]. Widely available hypoxic devices (e.g., hypoxic gas generators) allow to rapidly simulate the hypoxic conditions of high altitude so that athletes acutely experience exercise in hypoxia at sea level and without any acclimatization. Acute hypoxia (AH) triggers various cardiovascular changes that challenge the cardiovascular system, especially during exercise. Specifically, enhanced sympathetic tone and pulmonary artery vasoconstriction occur during AH [5,6,7].

Whether exercise in AH exerts deleterious effects on cardiovascular function during exercise remains controversial. It appears to be well established that exercise performance is impaired during exercise in moderate hypoxia, i.e., when the oxygen fraction of inspired air (FiO_2_) is reduced to approximately 0.13–0.15%. The hypoxia-induced decline in exercise performance is due to a decrease in peripheral arterial oxygen saturation (SatO_2_), an increase in pulmonary ventilation (VE), and an increase in afferent feedback from muscles that results in a downregulation of motor output from the central nervous system [8,9,10,11]. Moreover, hypoxia may activate the nitric oxide synthase, thereby depressing force generation of the diaphragm [12,13], and this may further contribute to the perception of fatigue and to challenge the cardiovascular system.

It remains unclear, however, if the cardiovascular system adequately increases cardiac output (CO) to meet the metabolic needs of the working muscle during AH. Specifically, one contention is whether cardiac performance is negatively affected by exercise in AH. While it is generally accepted that stroke volume (SV) is well preserved [6,7,8], whether cardiac filling and emptying are affected during exercise in AH remains controversial. Studies employing acute, normobaric hypoxia during rest and exercise reported an increase in left-ventricular deformation magnitude, which was attributed to hypoxia-induced systemic vasodilation and/or a sympathetic-mediated increase in myocardial contractility [14,15]. Furthermore, it has been recently reported that a brief bout of mild-intensity exercise in AH does not impair systolic or diastolic function. Instead, SV was well preserved owing to improvements in myocardial contractility and early diastolic function [16].

On the other hand, it has been shown that metaboreflex activation following exercise bouts in AH results in a decrease in SV, a phenomenon attributed to the capacity of AH to impair cardiac filling rate [17,18]. Moreover, the capacity to increase cardiac preload in response to exercise is decreased after only a few days at high altitude, potentially due to a reduction in plasma volume, impaired venous return, and/or impaired diastolic relaxation [7]. Collectively, these observations suggest that cardiac preload may be negatively affected by AH.

Thus, previous findings suggest that hemodynamic responses during exercise in AH result from a complex interplay between cardiac performance, which appears to be enhanced, and diastolic function, which, conversely, appears to be impaired. Taken together, it can be speculated that during exercise in AH, the impairment in diastolic filling is counterbalanced by the enhancement in systolic function, thereby sustaining SV. To the best of our knowledge, there are no studies, to date, that have investigated this possibility.

In the present investigation, we assessed the hemodynamic changes during a brief bout of acute, mild dynamic exercise during moderate acute normobaric hypoxia in young, healthy physically active males. We hypothesized that exercise in AH would impair diastolic function and reduce ventricular filling rate, but that these effects would be counteracted by enhanced systolic function and increased ventricular emptying rate, thereby sustaining SV.

## 2. Materials and Methods

### 2.1. Participants

The required sample size was determined with an online sample size calculator (https://www.stat.ubc.ca/~rollin/stats/ssize/n2.html, accessed on 10 June 2021). The criteria set to calculate the sample size were: (1) a power of 85%, (2) an overall type 1 error of 0.05, (3) an SD of 20%, and (4) a 20% difference between conditions in the studied variables, i.e., ventricular filling rate (VFR) and ventricular emptying rate (VER). Eighteen subjects were required to obtain adequate statistical power.

Twenty-five healthy Caucasian males between the age of 20 and 40 years were recruited to participate in the study. Smokers and individuals taking medications for any disease were excluded. Three participants did not complete the protocol as they experienced unbearable shortness of breath during the HYPO test, thus they were excluded from results, which included the remaining 22 subjects. The average (with 95% confidence interval, CI) of the participants’ age, body mass, and height were 31.1 (28.1–34.2) y, 73.1 (69.2–77.0) kg, and 175.8 (173.4–178.4) cm, respectively. All participants, as self-reported, were regularly involved in leisurely sport activities (i.e., amateur cycling and running) for at least four times/week, with an average of 8 ± 1.5 h/week. Participants were unaware of the nature of the study and were asked to refrain from alcoholic beverages and caffeine for at least 24 h before the experimental sessions.

The study was performed according to the Declaration of Helsinki and was approved by the ethics committee of the University of Cagliari (ref: letter n° 0120073832/30/03/2021). Written informed consent was obtained from all participants.

### 2.2. Experimental Protocol

#### Preliminary Test

All participants underwent a preliminary medical examination to assess their health status and to exclude cardiovascular or respiratory diseases followed by a cardiopulmonary exercise stress test (CPT) with ECG recording on a mechanically-braked cycle ergometer (Monark 828E, Vansbro, Sweden). The CPT consisted of a linear increase of workload (30 W/min), starting at 30 W, at a pedaling frequency of 60 rpm, until exhaustion, which was taken as the point at which the subject was unable to maintain a pedaling rate of at least 50 rpm. Maximum workload (W_max_) and maximum oxygen uptake (VO_2max_) were measured. Achievement of VO_2max_ was considered as the attainment of at least two of the following criteria: (1) a plateau in oxygen uptake (VO_2_) despite increasing workload (<80 mL·min^−1^), (2) a respiratory exchange ratio (RER) above 1.10, and (3) a heart rate (HR) ± 10 beats·min^−1^ of predicted maximum HR calculated as 220—age [19]. VO_2max_ was calculated as the average VO_2_ during the final 30 s of the incremental test. VO_2_, carbon dioxide production (VCO_2_), and VE were measured with a gas analyzer (ULTIMA CPX, MedGraphics, St. Paul, MN, USA) and calibrated immediately before each test as indicated by the manufacturer. During the preliminary test, participants familiarized themselves with the laboratory staff and equipment, thus allowing habituation to the environment and the cycle ergometer that was employed during the successive experimental sessions.

### 2.3. Exercise Test to Study the Hemodynamic Response in Normoxia and Hypoxia

After the preliminary visit and the CPT (interval 4–7 days), participants underwent two constant-load exercise tests to study their hemodynamics in normoxia (NORMO) and in hypoxia (HYPO). The NORMO and HYPO tests were randomly assigned to avoid any order effect and separated by at least 7 days (interval 7–10 days) [20]. Randomization was obtained using an online random sequence generator (https://www.random.org/sequences/, accessed on 13 July 2021). During both NORMO and HYPO tests, participants were fitted with a mask connected to a hypoxic gas generator (Everest Summit II Generator, Hypoxico, New York, NY, USA), which can provide a gas mixture with a reduced oxygen content that can be regulated. An FiO_2_ of 21% (corresponding to that of sea level) and 13.5% (corresponding to an altitude of ~3500 m) were delivered to the participants during the NORMO and the HYPO tests, respectively. The level of hypoxia in the HYPO test was chosen considering previous studies that demonstrated a significant reduction in exercise performance with this experimental setting [8,9,10]. Moreover, a similar experimental setting, with the same mild workload, was previously employed in our laboratory in recent investigations dealing with the cardiovascular effects of hypoxia during the recovery period from exercise in AH, and it was demonstrated that this setting was able to significantly reduce SatO_2_ and Cox [16,17,18]. Throughout the tests, FiO_2_ was constantly checked by an operator using an oxygen analyzer provided with the device (Maxtec, Handi+, Salt Lake City, UT, USA) and the participants were blinded about the content of oxygen they were breathing. In detail, participants were connected to a hypoxic gas generator and sat on the cycle ergometer for 4 min of rest before pedaling for 3 min against a workload corresponding to 30% of the W_max_ reached during the CPT, i.e., 76.14 W (CI: 70.3–81.9 W). A recovery period of 6 min was allowed.

### 2.4. Hemodynamic Assessment

Throughout the NORMO and HYPO tests, hemodynamic data were gathered using the technique of impedance cardiography, which uses changes in the thoracic impedance (Z_0_) to calculate SV. This method assumes that changes in SV are proportional to changes in Z_0_, measured with a low-amplitude, alternate electrical current applied to the thorax. Since electricity follows the ways of less impedance, the electrical current mainly flows along the great vessels in the mediastinum (i.e., aorta and vena cavae) so that the volume of blood inside the aorta is the major determinant of Z_0_. It follows that changes in Z_0_ reflect changes in blood volume inside the aorta, which in turn depends on SV, which can be calculated employing standard formulas.

Participants were connected to an impedance cardiograph (NCCOM 3, BoMed Inc., Irvine, CA, USA) which provided traces of Z_0_ and ECG, that were converted in digital signal and stored with a digital recorder (ADInstruments, PowerLab 8sp, Castle Hill, Australia). The sample rate was 500 Hz. ECG, Z_0_, and its first derivative (dZ/dt) were then analyzed offline to calculate HR (calculated as the reciprocal of R–R intervals), the pre-ejection period (PEP), and the ventricular ejection time (VET). PEP is the time spent by the left ventricle developing the amount of pressure necessary to overcome aortic pressure. This time interval is inversely related to sympathetic activity but is not influenced by parasympathetic tone; thus, we utilized it as an index inversely related to sympathetic tone [21]. PEP was calculated as the time interval between the onset of the QRS wave on the ECG and the beginning of the systolic deflection on the dZ/dt trace, while VET was calculated as the time interval between the systolic deflection of dZ/dt and the local minimum of dZ/dt measured in the same cardiac cycle [22,23,24].

SV was calculated employing the Bernstein’s formula [25]. We further determined diastolic time (DT), which was calculated as the difference between the duration of the R–R interval and the sum of PEP and VET. To obtain a measure of VFR, a measure of diastolic function and venous return, the ratio of SV and DT was calculated [23,26,27,28]. Moreover, VER, a measure of cardiac systolic performance, was calculated as the ratio of SV and VET [26,27,28].

Cardiac output (CO) was obtained as the product of SV and HR, while systolic (SBP) and diastolic blood pressure (DBP) were assessed with a manual sphygmomanometer applied on the non-dominant arm by the same physician throughout all experimental sessions to avoid any operator-dependent bias. Mean arterial pressure (MAP) was derived from SBP and DBP, utilizing a formula that corrects the MAP measure taking into consideration changes in DT and systolic time during exercise-induced tachycardia [29]. Systemic vascular resistance (SVR) was obtained with the ratio of MAP and CO multiplied by 80, where 80 is a conversion factor applied to have standard resistance units.

To verify whether the hypoxic stimulus was effective, SatO_2_ was continuously measured through finger pulse oxymetry (Nonin, SenSmart X-100, Plymouth, MN, USA). The same device was employed to assess cerebral oxygenation (Cox) with the near-infrared spectroscopy (NIRS) method. Two probes were positioned on the left and the right sides of the forehead over the ipsilateral eyebrow. Sensors were taped and covered with a headband to keep the probe in a fixed position and prevent outer light from interfering with NIRS measurements. Care was taken to avoid that the headband did not cause any blood flow occlusion.

Figure 1 is a schematic representation of the experimental protocol.

### 2.5. Data Analysis

Data are presented as mean with CI. The Kolmogorov–Smirnov test was utilized to verify the normality of the variables. Since all variables were normally distributed, parametric analysis was employed. Hemodynamic values were analyzed at the fourth minute of rest and at the third minute of exercise (i.e., when a steady state in cardiovascular parameters was expected to be reached) [17,18], and at the third and sixth minutes of recovery to have an identical timing with respect to previous timepoints.

Data from each subject were averaged over 1 min; thus, each time point of the mean group data represents the average of 22 datapoints, i.e., one for each participant. Differences in studied variables were carried out using a two-way analysis of variance (ANOVA) (factors of time and condition: NORMO and HYPO) followed by Bonferroni post hoc when appropriate. Statistical analysis was performed using commercially available software (GraphPad Prism). A *p* value < 0.05 was considered to determine statistical significance.

## 3. Results

Results of the CPT are reported in Table 1.

Figure 2 shows the time course of SatO_2_ (panel A) and Cox (panel B) during the NORMO and HYPO tests, respectively. The HYPO test induced a significant reduction in SatO_2_ with respect to the NORMO test throughout the protocol sessions. During the exercise phase of the HYPO test, the reduction in SatO_2_ was significantly lower than during rest and recovery of the same test (96.6; CI: 95.14–98.06 vs. 83.08%; CI: 80.97–85.03 for the NORMO and HYPO tests, respectively). Cox was lower during the exercise and the recovery phases of the HYPO test with respect to the NORMO test (71.0 A.U.; CI: 68.07–73.93 vs. 62.8 A.U.; CI: 59.52–66.08 for the exercise phase of the NORMO and HYPO tests, respectively).

During the exercise phase of the HYPO and the NORMO tests, there was a similar increment in HR (129.5 bpm; CI: 120.8–138.2 vs. 135.4 bpm; CI: 127.9–142.9 for the NORMO and HYPO tests, respectively; Figure 3, panel A), without any detectable effect of condition. Similarly, SV increased during the exercise phase of both tests (99.1 mL; CI: 87.1–111.0 vs. 98.9 mL; CI: 88.7–109.1 for the NORMO and HYPO tests, respectively; Figure 3, panel B), without any condition effect. As a result of the parallel changes in HR and SV, CO was similar between the NORMO and HYPO tests throughout experiments (12.6 L·min^−1^; CI: 11.2–13.9 vs. 13.2 L·min^−1^; CI 12.0–14.3 during the exercise phase, respectively; panel C).

Figure 4 demonstrates that all the time variables of the cardiac cycle decreased during exercise, without any difference between tests. In detail, during the exercise phase of the NORMO and HYPO tests, the PEP level was 93.5 (CI: 81.9–105.1) vs. 90.1 ms (CI: 81.0–99.1) (panel A), the VET level was 178.3 ms (CI: 167.1–189.5) vs. 172.9 ms (CI: 166.5–184.8) (panel B), and the DT level was 364.3 ms (CI: 307.1–421.5) vs. 358.5 ms (303.3–413.7) (panel C).

Figure 5 shows that during the exercise phase of the NORMO and HYPO tests, VFR, VER, and MAP increased without any detectable effect of condition. In particular, VFR reached a value of 528.9 mL·s^−1^ (CI: 451.4–606.4) vs. 556.4 mL·s^−1^ (CI: 419.7–621.1) (panel A), VER a value of 558.9 mL·s^−1^ (CI: 497.2–620.6) vs. 577.3 mL·s^−1^ (CI: 516.8–637.8) (panel B), and MAP a value of 101.6 mmHg (CI: 101.0–112.2) vs. 102.7 mmHg (97.0–108.4) (panel C) for the NORMO and HYPO tests, respectively. Figure 5 (panel D) also shows that SVR decreased during the exercise phase of both tests, reaching a level of 677.0 dynes·s^−1^·cm^−5^ (CI: 669.5–684.4) vs. 648.8 dynes·s^−1^·cm^−5^ (CI: 679.6–718.8) in the NORMO and HYPO tests, respectively, without any condition effect.

## 4. Discussion

Exercise in hypoxia, such as hiking at altitude or during hypoxic training, may pose a challenge to body homeostasis. Moreover, hypoxia is the pathophysiological basis of several cardiovascular and respiratory diseases [6,30]. In recent years, exercise in AH has been increasingly utilized as an athletic training tool, although hemodynamic consequences, potential risks, and benefits of this kind of training have yet to be completely elucidated [3,31,32,33]. The present investigation aimed to study the hemodynamic effects of a bout of mild dynamic exercise in moderate normobaric AH in young, physically active males. Specifically, we assessed whether diastolic and/or systolic function were affected by this maneuverer as past research yielded conflicting results on whether cardiac filling and performance are enhanced or impaired by moderate AH during exercise [4,6,11,16,17,18]. As shown by Figure 2, the protocol employed successfully induced significant reductions in SatO_2_ and Cox, thereby testifying that participants experienced real AH during rest, exercise, and recovery. Then, there was an O_2_ unloading at tissue level and, likely, a right shift in the oxygen–dissociation curve of Hb, although we did not assess tissue temperature, pH, CO_2_ production, and 2,3-diphosphoglycerate, which are all well-known factors causing Hb O_2_ unloading [20].

Based on the results of the present study, our hypothesis that moderate AH affects hemodynamics during mild dynamic exercise in healthy, physically active males should be rejected. We found that neither VFR, which is a measure of diastolic flux, nor VER, which is a measure of cardiac performance, were significantly different between the NORMO and HYPO tests. Therefore, AH did not induce any significant changes in the rate of ventricular filling and emptying. Accordingly, we also found that SV, which depends on a balance between ventricular emptying and filling [26,27,28], was unaffected by AH.

HR was not any different between the two tests and, as a result, CO was similar between tests. These findings demonstrate that the pumping capacity of the heart was unchanged by AH. Furthermore, there were no significant adjustments in MAP or SVR in response to hypoxia. Similarly, none of the cardiac cycle times considered (i.e., PEP, VET, and DT) showed any variation during AH during rest, exercise, and recovery. Inasmuch as PEP is inversely related to sympathetic tone, there was not significant sympathetic activation in our experimental setting of AH.

Collectively, these data do not support any significant hemodynamic adjustment during mild dynamic exercise in moderate AH, notwithstanding the significant blood arterial hypoxia and the reduction in Cox. Our results are in accordance with data demonstrating that SV is normally maintained during exercise in moderate AH [6,7,11,16]. Moreover, this did not appear to be the consequence of any particular adjustment in hemodynamics, as none of the cardiovascular parameters assessed were any different between the NORMO and HYPO tests. Therefore, the present results can be interpretated as an absence of health issue due to hypoxia and a lack of substantial cardiovascular stimulus during exercise training in moderate hypoxia at mild intensity.

Interestingly, CO was not augmented in response to AH during exercise. Indeed, it should be expected that CO is augmented when hemoglobin saturation and O_2_ delivery are reduced, such as during hypoxia. This may suggest a failure of the cardiovascular system to adjust CO to meet the O_2_ needs of the working muscle during mild dynamic exercise in moderate acute AH. However, an explanation for this phenomenon is that O_2_ extraction at the muscular level successfully maintained O_2_ uptake despite the reduction in O_2_ delivery [34,35]. This is in accordance with the concept that the increase in CO during AH is a function of the severity of the imposed hypoxia, and that in moderate AH, CO is usually sufficient to preserve O_2_ delivery; however, this is not the case in severe hypoxia [6,36]. Yan and colleagues [11] reported similar levels of CO during two bouts of cycling in normoxia and hypoxia. Thus, our results support the concept that moderate normobaric AH during submaximal exercise does not pose a significant cardiovascular challenge and is usually well tolerated by healthy, active individuals. In this regard, it should also be considered that neither HR nor PEP, an index sensitive to changes in sympathetic nerve activity, were any different between the NORMO and the HYPO tests, thereby indicating that sympathoexcitation did not occur in our experimental setting.

Our results also negate the presence of significant vasodilation in response to AH as SVR was not any different between the NORMO and the HYPO tests. While exercise in hypoxia is known to cause the production of a variety of vasodilating metabolites, such as nitric oxide and prostaglandins [37,38], in our experimental setting, we did not observe any drop in SVR, thus indicating that the exercise bout was too short and/or too mild to induce any metabolite production able to induce significant vasodilation. Alternatively, it could be that hypoxic vasodilation was restrained by an increase in sympathetic tone. However, this occurrence was unlikely since, as previously mentioned, we did not observe any variation in HR or PEP, which are suggestive for sympathoexcitation. Moreover, MAP was not any different between conditions throughout the experimental protocols, and this suggests that that arterial baroreflex was not challenged, nor that baroreflex changed its control set point in response to the changes in SatO_2_ and O_2_ delivery [39].

A phenomenon that should be highlighted is that we found a significant reduction in Cox during the exercise bout in AH and the following recovery. It has been observed that hypoxia causes cognitive impairment even at mild levels [40], and that it exacerbates fatigue and may contribute to the decision to stop exercising [8,41]. The fact that three out of 25 participants terminated the protocol due to self-reported “unbearable shortness of breath” during the exercise bout in AH may be related to the drop in Cox which exacerbated their sensation of fatigue. Therefore, even though the results of our study did not reveal any significant cardiovascular challenge imposed by our protocol, it is important to note that SatO_2_ and Cox were significantly reduced, and this may be a concern in terms of cognitive functions and of the sensation of fatigue. Thus, although from our study it appears that this type of exercise is safe for a cardiovascular point of view, its consequences on cognition and the sensation of fatigue should not be neglected. Further investigation is warranted to better clarify the effect of exercise during normobaric AH on central nervous system function.

### Limitations of Our Study

One limit of the present investigation was the lack of any measures of cardiac volumes, which would have helped provide a more comprehensive understanding of any potential hemodynamics changes. Specifically, echocardiography permits the evaluation of end-diastolic and end-systolic volumes, and the indirect measure of pulmonary artery pressure, which is known to rapidly change in response to hypoxia. A second limitation was that the duration and/or intensity of the exercise bouts in hypoxia were potentially too short and/or mild to pose a significant challenge to the cardiovascular system. Although our protocol induced significant reductions in SatO_2_ and Cox, and three dropouts from the study, it is possible that longer periods in hypoxia with a more intense workload would have induced a more profound hemodynamic perturbation. In this regard, it has been often observed that hypoxia normally leads to tachycardia when compared with normoxia at the same absolute workload, although this is not unanimously reported since, along with sympathetic tone, parasympathetic tone can also increase during exercise in hypoxia [6,42,43]. It should also be noted that our experiments were conducted in a laboratory setting; thus, care should be taken if extrapolating data to real high-altitude exposure. Finally, results are applicable only in young, physically active males and not in females and/or patients. Only male subjects were included in the study to eliminate any potential effect of hormonal changes during the menstrual cycle that might affect vascular responsiveness and interfere with hemodynamic response.

## 5. Conclusions

The results of the present investigation do not support the hypothesis that a brief, mild exercise bout in acute moderate normobaric hypoxia induces significant changes in hemodynamics. Although we detected significant reductions in SatO_2_ and Cox, none of the cardiovascular variables considered were changed in response to exercise in hypoxia in comparison with exercise in normoxia. It then appears that the circulatory system well tolerates this particular type of hypoxic exercise, which does not pose any significant challenge for the cardiovascular system.

### Practical Applications

In this study, we demonstrated that a brief bout of mild exercise in hypoxia was unable to affect hemodynamics in healthy subjects. With the aim to identify a “threshold” of cardiovascular tolerance, we propose that future research should focus on exercise sessions with more sustained hypoxia, longer duration, or higher workloads. This would help to optimize hypoxia as a useful tool for training and to develop training sessions in hypoxia avoiding risks of early fatigue due to cardiovascular failure.

## Figures and Tables

**Figure 1 ijerph-19-04558-f001:**
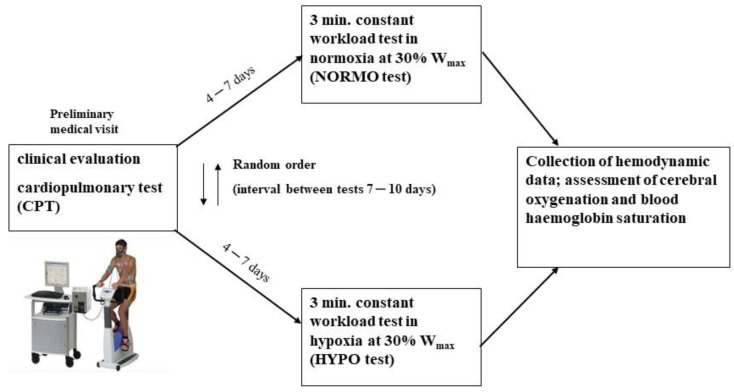
Schematic representation of the experimental protocol.

**Figure 2 ijerph-19-04558-f002:**
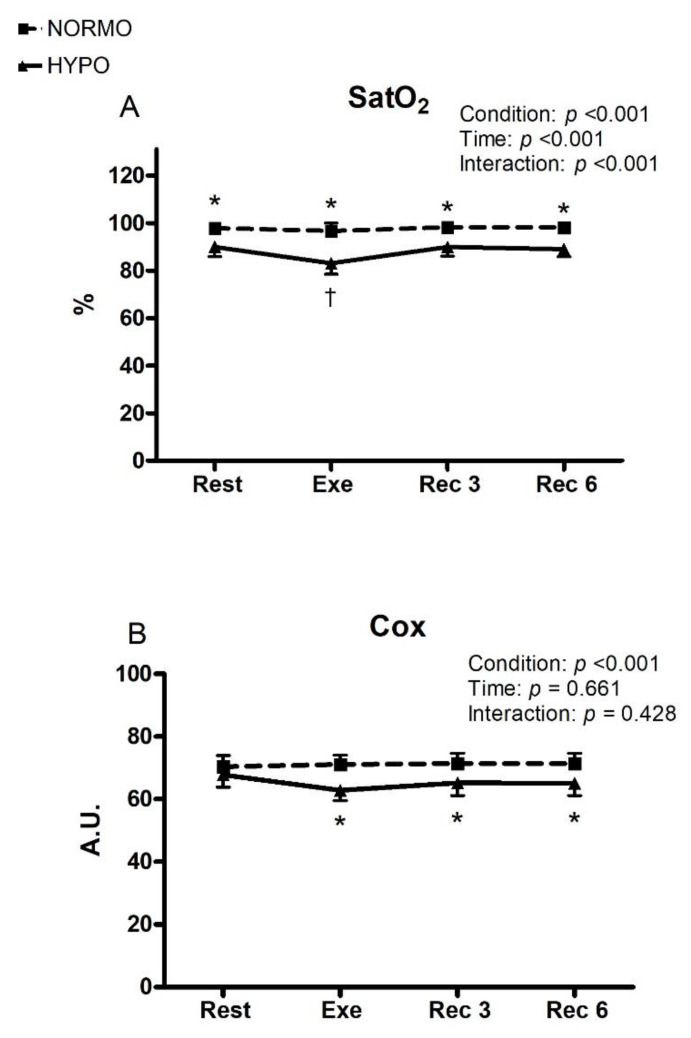
Changes in the level of peripheral blood O_2_ saturation (SatO_2_, (**panel A**) during the sessions of exercise (Exe) in normoxia (NORMO) and in normobaric hypoxia with a FiO_2_ of 13.5% (HYPO). **Panel B** shows changes in regional cerebral oxygenation (Cox) during the same tests. Values are mean with 95% confidence interval. N = 22. * = *p* < 0.05 between NORMO and HYPO at the same time point. † = *p* < 0.05 vs. rest, 3 min of recovery (Rec 3), and 6 min of recovery (Rec 6) of the same test.

**Figure 3 ijerph-19-04558-f003:**
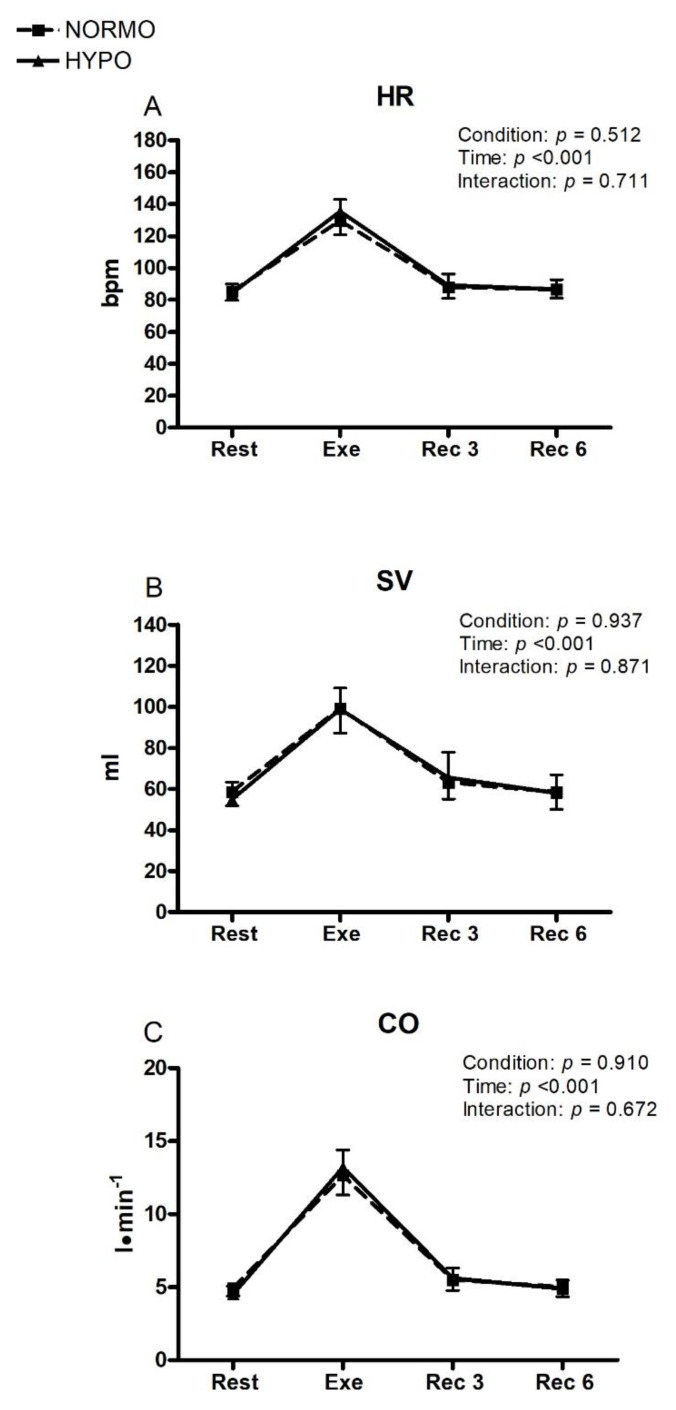
Changes in the level of heart rate (HR, **panel A**), stroke volume (SV, **panel B**), and cardiac output (CO, **panel C**) during the sessions of exercise in normoxia (NORMO) and in normobaric hypoxia with a FiO_2_ of 13.5% (HYPO). Values are mean with 95% confidence interval. *N* = 22.

**Figure 4 ijerph-19-04558-f004:**
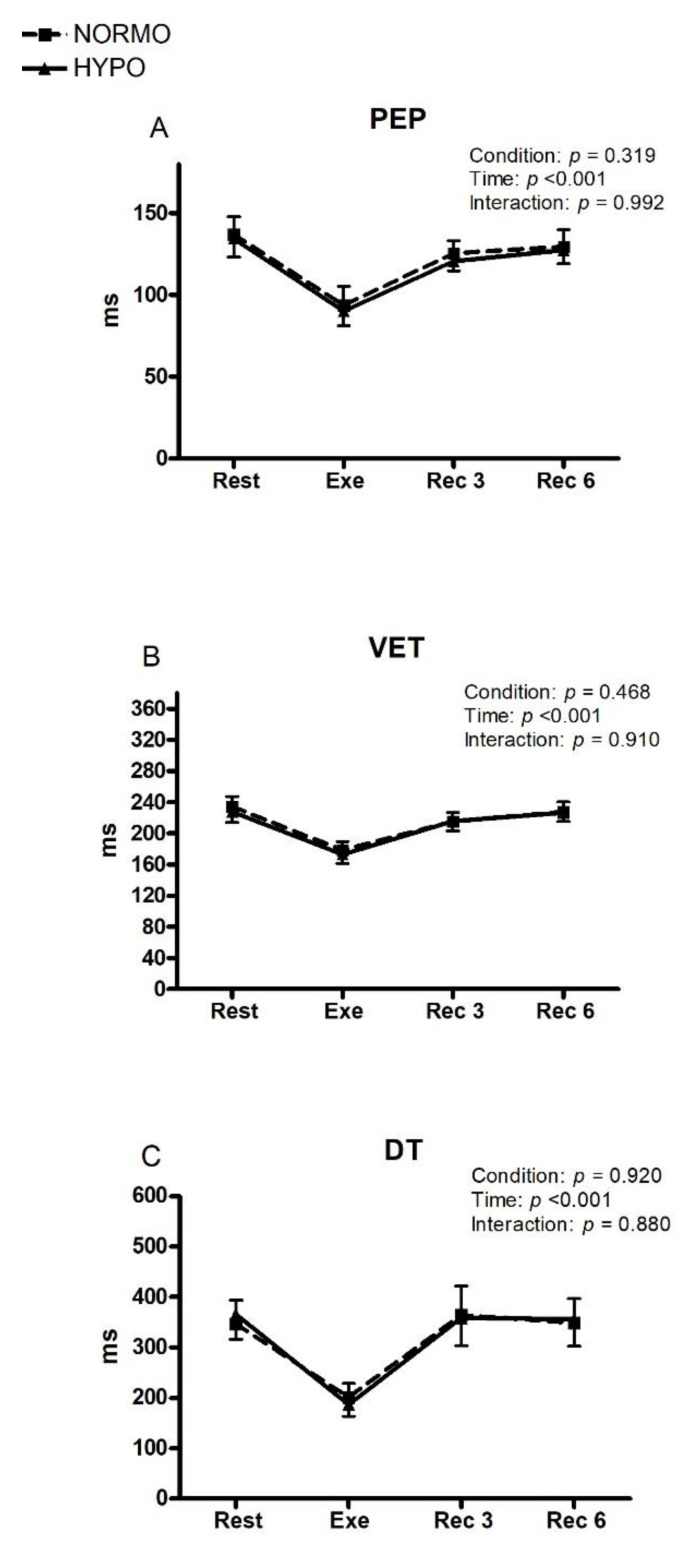
Changes in the level of pre-ejection period (PEP, **panel A**), ventricular ejection time (VET, **panel B**), and diastolic time (DT, **panel C**) during the sessions of exercise in normoxia (NORMO) and in normobaric hypoxia with a FiO_2_ of 13.5% (HYPO). Values are mean with 95% confidence interval. *N* = 22.

**Figure 5 ijerph-19-04558-f005:**
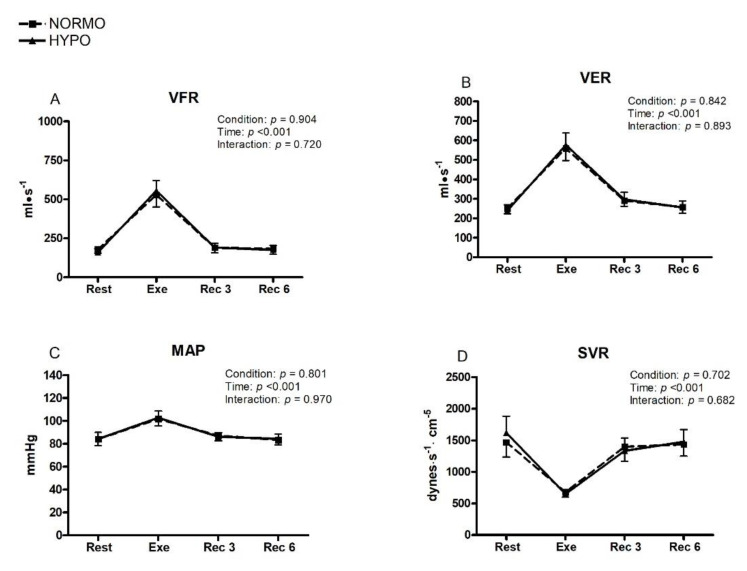
Changes in the level of ventricular filling rate (VFR, **panel A**), ventricular emptying rate (VER, **panel B**), mean arterial pressure (MAP, **panel C**), and systemic vascular resistance (SVR, **panel D**) during the sessions of exercise in normoxia (NORMO) and in normobaric hypoxia with a FiO_2_ of 13.5% (HYPO). Values are mean with 95% confidence interval. *N* = 22.

**Table 1 ijerph-19-04558-t001:** Mean with 95% confidence interval of metabolic data at maximum workload (W_max_) collected during the cardiopulmonary test. *N* = 22.

W_max_ (W)	253.8 (243.6–273.0)
VO_2max_ (mL·kg^−1^·min^−1^)	40.29 (38.48–42.1)
VO_2max_ (mL·min^−1^)	2939 (2759–3119)
VCO_2max_ (mL·min^−1^)	3504 (3289–3719)
RER_max_	1.19 (1.15–1.23)
V_Emax_ (L·min^−1^)	94.74 (87.57–101.90)
HR_max_ (bpm)	177.4 (172.9–181.9)

VO_2max_ = maximum oxygen uptake expressed indexed for body mass (second line) as well as in absolute values (third line); VCO_2max_ = maximum carbon dioxide production; RER_max_ = maximum respiratory exchange ratio; V_Emax_ = maximum pulmonary ventilation; HR_max_ = maximum heart rate.

## Data Availability

Dataset will be shown upon request.

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
