# Peer review of "Acute Exercise with Moderate Hypoxia Reduces Arterial Oxygen Saturation and Cerebral Oxygenation without Affecting Hemodynamics in Physically Active Males"

_ijerph, 2022, doi:10.3390/ijerph19084558_

Round 1
Reviewer 1 Report
Ln100 - “Written informed consent was obtained from all participants” – “…participants.” - A dot sign “.” needed at the end of the sentence.
Ln199-201 – “V.O2max = maximum oxygen uptake expressed indexed for body mass (second line) as well as in absolute values (third line); V.CO2max = maximum carbon dioxide production; RERmax = maximum respiratory exchange ratio; VEmax = maximum pulmonary ventilation; HRmax =maximum heart rate. N = 22.” - I suggest placing these abbreviations below the table.
This work is very informative and provides valuable findings on the current topic. The aim, discussion as well as study results are comprehensively presented, thus giving clear insight into this matter.
I suggest accepting this manuscript upon a minor correction mentioned above.
Author Response
To the editor and reviewers: please note that we are sending the revised version of the manuscript with track changes in red underlined, so that changes can be easily found.
Reviewer Comments:
Reviewer 1
We thank you the reviewer for her/his useful comments and appreciation upon our manuscript.
Comments and Suggestions for Authors
Ln100 - “Written informed consent was obtained from all participants” – “…participants.” - A dot sign “.” needed at the end of the sentence.
A dot sign has been added.
Ln199-201 – “V.O2max = maximum oxygen uptake expressed indexed for body mass (second line) as well as in absolute values (third line); V.CO2max = maximum carbon dioxide production; RERmax = maximum respiratory exchange ratio; VEmax = maximum pulmonary ventilation; HRmax =maximum heart rate. N = 22.” - I suggest placing these abbreviations below the table.
Abbreviations have been moved accordingly to reviewer’ suggestions.
This work is very informative and provides valuable findings on the current topic. The aim, discussion as well as study results are comprehensively presented, thus giving clear insight into this matter.
I suggest accepting this manuscript upon a minor correction mentioned above.
Reviewer 2 Report
The manuscript by Mulliri and colleagues describes a study on the investigation of hemodynamics responses during an acute moderate-intensity bout of dynamic exercise during normoxia or hypoxia conditions. I found that the rationale, the design setting and the methodology were sound and the results were well supported and described. I believe the paper warrants publication but there are a few issues I would like to see considered. Please find below some specific comments that I hope would contribute in improving the overall manuscript quality.
Introduction
Line 45: please consider to expand the paragraph addressing some notions regarding the role of respiratory muscles under normoxia and hypoxia, as well as, the contribution of Nitric Oxide.
Methods
Line 101: please consider to add an image reporting the timeline evolution of each experimental setting procedure.
Line 119: did participants executed a warm-up procedure just before the preliminary test? In case of affermative response, consider to briefly explain it.
Line 143: please, if it is possible, consider to add inter-reliability coefficients about this testing methods
Discussion
Line: 265: please consider to discuss the role of hemoglobin affinity during an hypoxic environment, the systemic increase of erythropoietin obtained in this condition and their consequences during physical exercise.
Author Response
To the editor and reviewers: please note that we are sending the revised version of the manuscript with track changes in red underlined, so that changes can be easily found.
Comments and Suggestions for Authors
The manuscript by Mulliri and colleagues describes a study on the investigation of hemodynamics responses during an acute moderate-intensity bout of dynamic exercise during normoxia or hypoxia conditions. I found that the rationale, the design setting and the methodology were sound and the results were well supported and described. I believe the paper warrants publication but there are a few issues I would like to see considered. Please find below some specific comments that I hope would contribute in improving the overall manuscript quality.
We thank you the reviewer for her/his useful comments and appreciation upon our manuscript.
Introduction
Line 45: please consider to expand the paragraph addressing some notions regarding the role of respiratory muscles under normoxia and hypoxia, as well as, the contribution of Nitric Oxide.
Some new references have been added to explain that NO may depress diaphragm contractility thereby further contribute to challenge the cardiovascular system.
Methods
Line 101: please consider to add an image reporting the timeline evolution of each experimental setting procedure.
A new picture (Figure 1) reporting the experimental setting has been added.
Line 119: did participants executed a warm-up procedure just before the preliminary test? In case of affermative response, consider to briefly explain it.
Response: they did not warm-up as the test started at very mild workloads (30W/min), which could be considered a sort of warm-up.
Line 143: please, if it is possible, consider to add inter-reliability coefficients about this testing methods.
This is for sure a good question, but, unfortunately, to date we do not have sufficient datasets to perform this kind of calculation.
Discussion
Line: 265: please consider to discuss the role of hemoglobin affinity during an hypoxic environment, the systemic increase of erythropoietin obtained in this condition and their consequences during physical exercise.
A sentence at the beginning of the “Discussion” has been added to address O2 affinity in the contest of our experimental set-up.
Reviewer 3 Report
The present study is of interest to investigate hemodynamics during an acute bout of mild, dynamic exercise during moderate normobaric acute hypoxia (AH).
Despite the interesting work, I strongly suggest following the comments to improve the quality of the manuscript.
Materials and methods
- I suggest the authors add a flowchart, to better elucidate the study design. In my opinion, will be more easiest for readers to follow all the rationale.
- Why do authors only assess male participants?
- L 123 - 124 "The NORMO and HYPO tests were randomly assigned and separated by at least 7 days (interval 7-10 days)." Please, explain why, using valid references to support your answer.
- Statistical procedures might need to be discussed using a within-subjects approach since basic group comparisons (i.e, NORMO vs. HYPO) were performed.
- It is not clear, the inferential test that was used to compare the difference between NORMO and HYPO (Figure 1, 2, 3 and 4).
- How was this comparison attempted? Did authors pool data for the comparison? How many data points were paired?
- Line 191-192 "Hemodynamic values were analyzed at the fourth minute of 191 rest, at the third minute of exercise, and at the third and sixth minutes of recovery."Please, explain better why using valid references to support your answer.
- Given the intra-individual variability, a within-subjects approach (appropriate for small samples) might be appropriate. Please see some recent works:
https://pubmed.ncbi.nlm.nih.gov/31527865/,
https://pubmed.ncbi.nlm.nih.gov/33672683/;
https://www.frontiersin.org/articles/10.3389/fphys.2021.678462/full.
- Figure 1, 2, 3 and 4: Y-axis, should start in 0.
- Authors should add 95 % IC in all findings.
- The authors should add the magnitude of differences between NORMO and HYPO, to better understand the Δ. And please, discuss all the magnitudes found.
5. Results, discussion and conclusion section should be rewritten accordingly to the previous commentaries.
6. Authors should also include a practical application section.
Author Response
To the editor and reviewers: please note that we are sending the revised version of the manuscript with track changes in red underlined, so that changes can be easily found.
Comments and Suggestions for Authors
The present study is of interest to investigate hemodynamics during an acute bout of mild, dynamic exercise during moderate normobaric acute hypoxia (AH).
Despite the interesting work, I strongly suggest following the comments to improve the quality of the manuscript.
We thank you the reviewer for her/his useful comments and appreciation upon our manuscript.
Materials and methods
I suggest the authors add a flowchart, to better elucidate the study design. In my opinion, will be more easiest for readers to follow all the rationale.
As also suggested by reviewer 2, a picture explaining the study set-up has been added.
Why do authors only assess male participants?
We chose to enroll only male subjects to eliminate any potential effect of hormonal changes during the menstrual cycle that might affect vascular responsiveness and interfere with the hemodynamic response. This is a limit already outlined in the previous version of the manuscript. The section has been now extended with this information.
L 123 - 124 "The NORMO and HYPO tests were randomly assigned and separated by at least 7 days (interval 7-10 days)." Please, explain why, using valid references to support your answer.
Randomization was necessary to prevent any order effect. Tests were separated by 7 days to allow recovery and to prevent the effect of any metabolic adaptation due to exercise session in hypoxia. A new reference has been added in support of our experimental design.
Statistical procedures might need to be discussed using a within-subjects approach since basic group comparisons (i.e, NORMO vs. HYPO) were performed. It is not clear, the inferential test that was used to compare the difference between NORMO and HYPO (Figure 1, 2, 3 and 4).
Thank you for your careful revision of the text as a sentence was actually missing in the previous version of our manuscript. It was the two-way ANOVA, the information has been added in the new version of the manuscript.
How was this comparison attempted? Did authors pool data for the comparison? How many data points were paired?
This information has been added in the new version of the manuscript.
Line 191-192 "Hemodynamic values were analyzed at the fourth minute of rest, at the third minute of exercise, and at the third and sixth minutes of recovery." Please, explain better why using valid references to support your answer.
We used the fourth of rest and the third minute of exercise as we expected that hemodynamic values were in a steady state condition at these timepoints. References have been used in support of our choice. Recovery timepoints were chosen to have an identical timing as previous timepoints.
Given the intra-individual variability, a within-subjects approach (appropriate for small samples) might be appropriate. Please see some recent works:
https://pubmed.ncbi.nlm.nih.gov/31527865/,
https://pubmed.ncbi.nlm.nih.gov/33672683/;
https://www.frontiersin.org/articles/10.3389/fphys.2021.678462/full.
In our opinion, this approach is not useful when standard statistics do not find out any difference between condition. Since we did not find any significant difference in hemodynamics between conditions, we prefer not to add any other data to the text, which is already cumbersome.
Figure 1, 2, 3 and 4: Y-axis, should start in 0.
Authors should add 95 % IC in all findings.
Although we found the request to add 95% CI unusual, figures have been redone with y-axes starting at 0 and with 95% CI instead of SD. Moreover, results have been re-written with 95% CI.
The authors should add the magnitude of differences between NORMO and HYPO, to better understand the Δ. And please, discuss all the magnitudes found.
As for the previous point, in our opinion, this approach is not useful since we did not find any significant difference in hemodynamics between the NORMO and the HYPO tests.
- Results, discussion and conclusion section should be rewritten accordingly to the previous commentaries.
As previously explained, we preferred not to change our statistical approach. Thus, these sections of the manuscript were not re-written, with the exception of Results, which has been rewritten with 95% CI instead of SD.
- Authors should also include a practical application section.
A practical application section has been added at the end of the discussion.
Round 2
Reviewer 3 Report
I am happy with the current version of the manuscript.
The authors did a good job on reviewing the manuscript and answering all the revisions maded.